# Evidence for a Classical Dissociation between Face and Object Recognition in Developmental Prosopagnosia

**DOI:** 10.3390/brainsci14010107

**Published:** 2024-01-22

**Authors:** Christian Gerlach, Randi Starrfelt

**Affiliations:** 1Department of Psychology, University of Southern Denmark, 5230 Odense, Denmark; 2Department of Psychology, University of Copenhagen, 1353 Copenhagen, Denmark; randi.starrfelt@psy.ku.dk

**Keywords:** developmental prosopagnosia, dissociation, object recognition, selectivity, visual similarity

## Abstract

It is still a matter of debate whether developmental prosopagnosia is a disorder selective to faces or whether object recognition is also affected. In a previous study, based on a small sample of developmental prosopagnosics (DPs; N = 10), we found impairments in both domains although the difficulties were most pronounced for faces. Importantly, impairments with faces and objects were systematically related. We suggested that that the seemingly disproportional impairment for faces in DP was likely to reflect differences between stimulus categories in visual similarity. Here, we aimed to replicate these findings in a larger, independent sample of DPs (N = 21) using the same experimental paradigms. Contrary to our previous results, we found no disproportional effect of visual similarity on performance with faces or objects in the new DP group when compared to controls (N = 21). The new DP group performed within the control range, and significantly better than the old DP-group, on sensitive and demanding object recognition tasks, and we can demonstrate a classical dissociation between face and object recognition at the group level. These findings are perhaps the strongest evidence yet presented for a face-specific deficit in developmental prosopagnosia.

## 1. Introduction

Developmental prosopagnosia (DP) is a disorder characterized by a severe lifelong impairment in the ability to learn and recognize faces that cannot be ascribed to neurological disorders, low-level visual deficits such as poor acuity, or intellectual reduction [1]. There is little consensus on the criteria for the classification of the disorder [1,2,3,4], and the estimates of its prevalence vary from 1 to 5%, depending on the specific criteria used for classification [5].

Even though several hundred cases of DP have been reported since its original description by McConachie in 1976 [6], there is still debate concerning which cognitive processes are impaired in DP and whether the disorder is even selective for faces [7]. In 2016, we reported findings from a small sample of DPs (N = 10) where we focused on examining their object recognition abilities [8]. The main finding was that while this group of DPs did show greater impairments with face than with object recognition, object recognition was also impaired. This pattern of impairments in two domains—of which one is significantly more pronounced than the other—corresponds to what is defined as a strong dissociation. In comparison, a classical dissociation is defined as a pattern with an abnormal performance in one domain and a within-normal-range performance in another (related) domain [9].

The finding of a strong rather than a classical dissociation between the domains of face and object recognition can be explained in two ways: (i) as two functionally unrelated deficits that tend to co-occur [10] or (ii) as impairments that reflect a single underlying mechanism that is differentially affecting the two domains. We preferred the latter explanation as we also found a systematic relationship between the degrees of face and object recognition deficits across the DPs such that the degree of object recognition difficulty predicted the degree of face recognition difficulty [8]. The question then is what kind of deficit that can impair performance in both domains but also cause greater problems with one domain (faces) relative to the other (objects). One obvious candidate is visual similarity that is high for faces but not necessarily for objects [11].

To examine this aspect, which has been neglected in the literature on DP [11], we tested the DPs performance with faces and objects when the degree of perceptual differentiation was kept constant across conditions and the degree of visual similarity was manipulated systematically. This was performed in a simultaneous matching paradigm by presenting stimulus pairs of either faces or houses where the participants had to decide whether the stimuli in a pair were identical or not. The pairs differed in either the second-order relations (e.g., difference in the spacing between the eyes) or in their constituent features (e.g., different shape of the nose). More importantly, differences along these dimensions varied parametrically so that the pairs differed in either one, two, three, or four respects or none at all (in which case they were identical) (see Figure 1).

The pattern of results was rather clearcut. By manipulating the degree of visual similarity between stimuli in each domain, we could demonstrate impairments either for faces (when visual similarity was the highest for faces) or houses (when visual similarity was the highest for houses) or equal performance across domains (when visual similarity was on par). Considered together, these findings suggest that differences in visual similarity could very well explain the category-effects that we observed in this group of DPs.

While these findings indicate that the previous reports of face selectivity may have been confounded by the difference in visual similarity between categories [11], they were found in a relatively small sample of DPs. Given that DP may be a heterogeneous disorder [12,13,14,15] in that it is not yet established whether face recognition difficulties occur for different reasons across individuals, it is important to examine whether these effects can be replicated in other samples. This is the main purpose of the present investigation where we used the simultaneous matching paradigm described above in a larger independent sample of individuals with DP (N = 21). In the following sections, we will refer to this larger sample as the *New-DP* group and the sample reported in Gerlach et al. [8,16] as the *Old-DP* group.

To anticipate the results, we were unable to replicate our previous findings as the New-DP group, in contrast to the Old-DP group, did not show impairments with both faces and houses when visual similarly was on par for the two categories. In fact, there was little evidence suggesting that the New-DP individuals differed from their matched controls in the simultaneous matching paradigm at all, despite their clear face recognition problems in other tasks. Thus, the New-DP group seems to have a much more selective deficit in face recognition than the Old-DP group, and one that cannot be explained in terms of category-differences in visual similarity. Therefore, to examine the degree of selectivity in the New-DP group, we also analyze and report the New-DP group’s performance on the object recognition tasks on which the Old-DP group was found to be most impaired. These analyses reveal that the New-DP group not only performs within the normal range of their matched controls but also significantly better than the Old-DP group. The present findings suggest that a classical dissociation can be found between face and object recognition in DP even when object recognition is tested in quite demanding conditions and when effects of visual similarity are unlikely to account for the category differences.

## 2. Materials and Methods

### 2.1. The New-DP Group and Their Matched Controls

The individuals in the New-DP group (N = 21) were selected from a larger group of self-referred individuals (N = 32) that we have tested over a period of several years, and whom we have classified as having DP as per the criteria described below. Eleven of these individuals did not complete the simultaneous matching paradigm, and thus they cannot provide data of relevance for the present replication attempt. Hence, the 21 participants presented here are all the new DPs that we have in our pool who completed the simultaneous matching paradigm. The sample is independent from the Old-DP group, as it was recruited at a later time.

All participants with DP completed structured interviews regarding everyday difficulty with facial identity recognition and possible family history of DP. They all reported severe difficulties with face recognition in their everyday life, as evaluated by the first part of the Faces and Emotion Questionnaire (FEQ, 29-items) [17]. The DP classification was ultimately based on abnormal scores in both the FEQ and the CFMT [18], for more details see [19]. All DPs performed within the normal range (score of 32 or less) on The Autism-Spectrum Quotient [20] and did not receive remuneration for their participation.

The control group (N = 21) was selected from a larger group of participants who had completed the same test protocol as the DPs including the CFMT and the FEQ. They were each selected to match one of the New-DPs as closely as possible in terms of gender, age, and education. No controls from the previous study were included, i.e., the sample was independent. All controls performed within the normal range (not below two SDs) on the CFMT, evaluated by the age- and gender-adjusted norms provided in Bowles et al. [21]. They also performed within the normal range on the Autism-Spectrum Quotient. Controls received gift certificates of ~120 DKK (~20 USD) per hour for their participation.

The New-DP group consisted of 15 females/6 males (age: M = 40, SD = 14; years of education: M = 16, SD = 1), and the control group consisted of 15 females/6 males (age: M = 40, SD = 13; years of education: M = 16, SD = 1.1). All participants were Caucasian Danish citizens.

### 2.2. The Simultaneous Matching Paradigm

The simultaneous matching experiment was originally developed by Collins et al. [22]. In the paradigm, participants are presented with stimulus pairs of either two faces or two houses and have to decide whether the pairs are identical or not. The stimulus pairs were arranged such that the two faces/houses were presented one above the other. Each stimulus in a pair subtended 2.5–5.3° of visual angle, and each pair was shown until the participant responded or for a maximum duration of 10 s. If no response was registered within 10 s, the trial was terminated and counted as an error. The pairs differed in either the second-order relations or in the features themselves, and differences in these dimensions varied parametrically so that the pairs either differed in one, two, three, or four respects or none at all (in which case, they were identical, see Figure 1). The presentation order was randomized across category and second-order and featural conditions. For additional information regarding the paradigm, see [8].

To replicate the findings reported by Gerlach et al. [8,16], we tested four hypotheses that are all derived from the basic assumption that it is a difference in visual similarity that is driving the category-specific effect observed for face processing.

**Hypothesis 1:** 
*The New-DP group will perform worse than controls with both faces and houses when the visual similarity among members of the two categories is similar. To examine this, we computed the average performance with faces and houses across the four similarity levels (0, 1, 2, and 3). This was performed separately for the second-order and the featural conditions. These averages were subjected to a mixed factorial ANOVA with Group (New-DP group vs. control group) as the between-subject factor and Category (faces vs. houses) as the within-subject factor.*


**Hypothesis 2:** 
*The New-DP group will be equally impaired with faces and houses relative to the controls when visual similarity is similar for the two categories, that is, there will be no significant interaction between Group and Category when visual similarity is the same across categories. A lack of significant interaction between Group and Category is of course negative evidence. Hence, to provide positive evidence, we also tested the two additional hypotheses below.*


**Hypothesis 3:** 
*Face processing will be more impaired than the processing of houses in the New-DP group when visual similarity is higher for faces than for houses. This was examined by comparing performance when the similarity level was the highest for faces (level 3) and the lowest for houses (level 0). Again, this was done in a mixed factorial ANOVA with Group and Category as factors conducted separately for the second-order and the featural conditions.*


**Hypothesis 4:** 
*The processing of houses will be more impaired than the processing of faces in the New-DP group when visual similarity is higher for houses than for faces. This was examined by comparing performance when the similarity level was the highest for houses (level 3) and the lowest for faces (level 0). Again, this was done in a mixed factorial ANOVA with Group and Category as factors conducted separately for the second-order and the featural conditions.*


In Gerlach et al. [8], these hypotheses were tested based on accuracy (percentage correct classifications in trials where the stimulus pairs differed; different-trials) rather than RTs. This was because too many of the Old-DPs had very high error rates in the paradigm, making group comparisons based on RT questionable. As the purpose of the present study is to replicate the original findings by Gerlach et al. [8], we also place the main weight on analyses of accuracy. We also note that even though visual similarity was varied parametrically in this paradigm, this does not mean that visual similarity was fully controlled. Even if a house pair only differed in one dimension and a face pair also differed in only one dimension, this does not guarantee that the house pair is just as visually similar as the face pair. It might be that the difference between the stimuli in the house pair (e.g., a difference between two windows) is perceived as more subtle than the difference between the stimuli in the face pair (e.g., a difference between two noses). Similarly, if a face pair differs in two dimensions rather than in one dimension, this does not mean that the visual similarity in one pair is twice that of the other. Nevertheless, by controlling the number of dimensions in which the pairs differ, we at least had a procedure for scaling the overall similarity between the stimulus pairs

## 3. Results

### 3.1. Simultaneous Matching

Hypotheses 1 and 2 (the same degree of similarity for faces and houses): For the second-order relation conditions, there was a main effect of Category (*F* (1,40) = 32.9, *MSe* = 2958, partial η^2^ = 0.45, *p* < 0.001), with more accurate performance for faces than for houses. Neither the main effect of Group or the interaction between Group and Category was significant (*p* = 0.6 and *p* = 0.23, respectively) (see Figure 2A). For the featural conditions, there was a main effect of Category (*F* (1,40) = 68.7, *MSe* = 1413, partial η^2^ = 0.63, *p* < 0.001), with more accurate performance for faces than for houses. Neither the main effect of Group nor the interaction between Group and Category was significant (*p* = 0.6 and *p* = 0.5, respectively) (see Figure 2B).

Hypothesis 3 (similarity higher for faces than houses): For the second-order relation conditions, there was a main effect of Category (*F* (1,40) = 28, *MSe* = 3269, partial η^2^ = 0.41, *p* < 0.001), with more accurate performance for houses than for faces. Neither the main effect of Group nor the interaction between Group and Category was significant (*p* = 0.63 and *p* = 0.17, respectively) (see Figure 2C). For the featural conditions, there was a main effect of Category (*F* (1,40) = 45.6, *MSe* = 5684, partial η^2^ = 0.52, *p* < 0.001), with more accurate performance for houses than for faces. Neither the main effect of Group nor the interaction between Group and Category was significant (*p* = 0.95 and *p* = 0.81, respectively) (see Figure 2D).

Hypothesis 4 (similarity higher for houses than faces): For the second-order relation conditions, there was a main effect of Category (*F* (1,40) = 217, *MSe* = 41,318, partial η^2^ = 0.84, *p* < 0.001), with more accurate performance for faces than for houses. Neither the main effect of Group nor the interaction between Group and Category was significant (*p* = 0.43 and *p* = 0.49, respectively) (see Figure 2E). For the featural conditions, there was a main effect of Category (*F* (1,40) = 358, *MSe* = 30,514, partial η^2^ = 0.9, *p* < 0.001), with more accurate performance for houses than for faces. Neither the main effect of Group nor the interaction between Group and Category was significant (*p* = 0.95 and *p* = 0.3, respectively) (see Figure 2F).

#### Interim Discussion

The results of the comparison between the New-DP group and the control group were clear: There was no evidence suggesting that the New-DPs were impaired relative to the control group regardless of the comparison. This was the case even when the similarity level within the stimulus pairs was at its highest for faces (similarity level = 3); a comparison that should have revealed a difference if the face deficit in the New-DP group was driven by visual similarity. This clearly departs from our previous findings where the Old-DP group performed more poorly than controls with both faces and houses, regardless of whether the stimuli differed in second-order relations or features.

The most surprising difference between the New-DPs and the Old-DPs is that the New-DPs did not show any impairment with faces in the simultaneous matching paradigm, not even when the similarity level was at its highest and ceiling effects cannot have prevented potential differences from emerging. One explanation for the finding that the New-DPs did not differ from the controls in this paradigm might be that the DPs were able to compare the stimuli in a pair by comparing features and relations between features in a sequential manner rather than simultaneously as a whole. By ‘sequential’ we mean that one can look for differences in a stimulus pair by concentrating on one feature (or second-order relation) at a time. This is possible because the stimuli in a pair are presented together. Such a strategy is not feasible in other paradigms, such as the CFMT, where stimuli are presented with a delay in between them. Hence, if DPs are using such a piecemeal strategy, we would expect RTs to be longer for the New-DP group than for the controls in all four conditions.

### 3.2. Examination of Speed–Accuracy Trade-Off

To examine if the New-DP group was trading higher accuracy for prolonged latency, we computed the mean correct reaction time (RT) for different-trials across the four similarity levels. Next, we performed a mixed factorial ANOVA on the mean correct RTs with Group (New-DP group vs. controls) as a between-subject factor and Condition (faces second-order relations, faces featural, houses second-order relations, and houses featural) as the within-subject factor. This analysis revealed a main effect of Condition (*F*(3,120) = 30.9, *MSe* = 4555049, partial η^2^ = 0.44, *p* < 0.001), and an interaction between Condition and Group (*F*(3,120) = 5.5, *MSe* = 813,008, partial η^2^ = 0.12, *p* = 0.001) (see Figure 3). The main effect of Group was not significant (*p* = 0.21). The interaction between Condition and Group was examined with analyses of the simple main effects. This revealed that the New-DP group and the controls differed significantly in the condition with the matching of second-order relations for faces (*t*_40_ = 2.77, d = 0.85, *p* < 0.01), with the DPs being slower than the controls. No effect of group was significant in the three other conditions (all *p*’s > 0.22).

#### Interim Discussion

As the New-DPs did not perform slower than the controls in general, it does not seem likely that they adopted a more sequential or piecemeal matching strategy than the controls. What remains to be explained is why the New-DP group performed well in the simultaneous matching paradigm relative to the Old-DP group when the good performance is unlikely to reflect a piecemeal strategy and they also—like the Old-DP group—exhibited clear face recognition difficulties when assessed using the CFMT, which is also based on accuracy, and the FEQ. A possible explanation is that the simultaneous matching paradigm does not tax memory (or learning) to any great extent. Consequently, impaired performance in the simultaneous matching paradigm is likely to reflect a perceptual problem in building up an adequate representation of the stimulus (an encoding deficit). If so, it would imply that the Old-DP group has a more severe perceptual deficit than the New-DP group. This explanation is also able to account for the observation that the Old-DP group was more impaired in second-order relations when visual similarity increased and more importantly was impaired for both faces and houses [8], which was clearly not the case for the New-DP group.

### 3.3. Direct Comparisons on Simultaneous Matching

To test this proposition directly, we compared the performance between the two DP-groups in the simultaneous matching paradigm by means of a mixed factorial ANOVA with Group (Old-DP group vs. New-DP group) as a between-subject factor andpercentage correct for each Condition (faces second-order relations, faces featural, houses second-order relations, and houses featural) as the within-subject factor. The sample size of the Old-DP group was nine because one of them did not perform this experiment.

This analysis revealed a main effect of Condition (*F*(1.63,45.67) = 23.4, *MSe* = 2384, partial η^2^ = 0.46, *p* < 0.001) and a main effect of Group (*F*(1,28) = 9.8, *MSe* = 8952, partial η^2^ = 0.26, *p* < 0.01). The interaction between Group and Condition was not significant (*p* = 0.07) (see Figure 4).

The finding that the New-DP group performed significantly more accurately across all conditions compared with the Old-DP group supports the proposition that the Old-DP group has a more severe perceptual deficit than the New-DP group.

### 3.4. Direct Comparisons on Object Recognition Tasks

If the Old-DP group does indeed have a more severe perceptual deficit than the New-DP group, and the deficit is *not* selective to faces as we have argued elsewhere [8,16], one would also expect the Old-DP group to perform worse than the New-DP group on tests of object recognition. To examine this, we compared the Old-DP group and the New-DP group on two tests of object recognition where we have previously found the Old-DP group to be impaired, i.e., the Cambridge Car Memory Task (CCMT) [23] and an object decision task with silhouettes [8]. The sample size of the Old-DP group was nine for the CCMT and eight for the object decision task with silhouettes, see [8].

The CCMT is identical to the CFMT in format and only differs in that cars are used as stimuli instead of faces. Performance in this task is measured in terms of accuracy (with 72 as the highest score).

In the object decision task, the participants are presented with 80 stimuli that depict real objects and 80 stimuli that depict nonobjects. The task is to decide whether stimuli represent real objects or nonobjects. The nonobjects were composed by exchanging single parts belonging to different real objects. This makes the task quite demanding as the decision cannot be made based on the identification of a single recognizable feature but must be made by integrating features across the whole stimulus.

In agreement with Gerlach et al. [8], the performance on the object decision task was assessed by means of A, which is a bias-free measure of sensitivity similar to A′ and A″ but based on a corrected formula by Zhang and Mueller [24]. The measure varies between 0.5 and 1.0 with higher scores indicating a better discrimination between objects and nonobjects.

#### 3.4.1. Results

The comparison of the Old-DP and the New-DP groups on these object recognition measures revealed that the Old-DP group was more impaired than the New-DP group on both the CCMT (*t*_28_ = 2.63, *d* = 1.05, *p* < 0.02) and the object decision task with silhouettes (*t*_27_ = 4.09, *d* = 1.7, *p* < 0.001). For completeness, we also compared the performance of the New-DP group with their controls on both measures and found no significant difference (*p* = 0.88 and *p* = 0.72 for object decision with silhouettes and the CCMT, respectively).

As mentioned in the introduction, some of the strongest support for the suggestion that the face and object recognition abilities were linked in the Old-DP group was the finding that object decisions with silhouettes correlated with the face recognition performance as indexed by the CFMT for the DPs (*r* = 0.87, *p* = 0.005, 95% CI = [0.55, 1]) but not for their controls (*r* = 0.07, *p* = 0.8). Even though the New-DP group did not differ significantly from their controls on the object decision task with silhouettes, it is still possible that a similar systematic association between face and object recognition performance might be present in the New-DP group. To examine this, we performed a correlation analysis between object decision performance with silhouettes and performance on the CFMT in the New-Group. This analysis did not reveal any systematic relationship (*r* = −0.06, *p* = 0.78, 95% CI = [−0.48, 0.38], BF_01_ = 5.8) (see Figure 5).

##### Interim Discussion

The analyses comparing performance on within-class object recognition (the CCMT), and object decision with silhouettes support the assumption that the New-DP group, in contrast to the Old-DP group, does not have a general perceptual deficit. Moreover, from what has been presented so far, it also seems as if the New-DP group, as opposed to the Old-DP group, has a specific problem with faces. However, it must be acknowledged that this conclusion so far is based on a mixture of positive evidence (abnormal performance on the CFMT) and negative evidence (within normal range performance on the CCMT and the object decision task with silhouettes). As we and others have argued, more firm support in favor of a disproportional deficit for faces relative objects should rest on positive evidence [25,26].

### 3.5. Evidence for a Disproportional Deficit for Faces Relative to Objects in the New-DP Group

To test for a disproportional impairment with faces relative to objects in the New-DP group, we applied the test developed by Crawford et al. [27] for identifying differential (disproportional) deficits. With this test, it is estimated whether the correlation between the dichotomous variable of group membership (here, New-DPs vs. controls) and the performance on test *X* (here, the CFMT) is significantly different from the correlation between the dichotomous variable of group membership and the performance on test *Y* (here object decision with silhouettes/the CCMT). To estimate this, the correlation between tasks *X* and *Y* is also taken into consideration. Hence, this test examines whether the difference in performance between tasks *X* and *Y* is larger in the New-DP group than it is in the control group. The test for a differential deficit was applied to both the CFMT vs. the CCMT and the CFMT vs. the object decision task with silhouettes.

#### 3.5.1. Results

The New-DP group exhibited a differential deficit on the CFMT relative to the CCMT (*t*_39_ = 5.58, *p* < 0.0001; CFMT and group membership correlation, *r* = 0.846; CCMT and group membership correlation, *r* = 0.056; overall CFMT and CCMT correlation, *r* = 0.091, *N* = 42), and also on the CFMT relative to the object decision task with silhouettes (*t*_39_ = 5.58, *p* < 0.0001; CFMT and group membership correlation, *r* = 0.846; object decision and group membership correlation, *r* = 0.024; overall CFMT and object decision correlation, *r* = 0.043, *N* = 42).

The findings that the New-DP group were (i) impaired on the CFMT, (ii) performed within normal range on the CCMT and the object decision task with silhouettes, and (iii) exhibited differences between the CFMT and the CCMT/object decision with silhouettes that were significantly different from those of the control group conforms to putative classical dissociations [25].

### 3.6. Examining Differences between the Old-DP and New-DP Groups in Face Recognition Severity

We have argued that the differences observed between the Old- and New-DP groups indicate that the Old-DP group has a general perceptual impairment whereas the New-DP group does not. An alternative explanation could be that the Old-DP group is simply more impaired than the New-DP group (also) with faces. In that case, the two groups do not differ in terms of the underlying cognitive dysfunction but merely in severity [3]. We tested this assumption by comparing the Old- and New-DP groups on the two measures of face recognition that were used to classify DP: the CFMT and the FEQ.

In addition, we tested whether the differences between the two groups on the CCMT or the object decision task with silhouettes were significantly larger than what one would expect given the differences between the two groups on the CFMT or the FEQ. This was performed by means of the test for a differential deficit [27]. These analyses provide an important supplement to the simple contrasts between the CFMT and the FEQ because they examine whether (potential) differences between the groups in the face recognition measures can account for (potential) differences the between the groups in the object recognition measures. For instance, if differential deficits are revealed, this will provide *positive* evidence that differences in face recognition performance between the groups cannot account for differences in object recognition performance between the groups.

#### 3.6.1. Results

The two groups differed significantly on the CFMT (*t*_28_ = 2.47, *p* = 0.02, *d* = 0.98, BF_01_ = 0.34, 95% credible interval [−0.16, 7.43]), with the Old-DP group being more impaired than the New-DP group, but not on the FEQ (*t*_28_ = −0.52, *p* = 0.62, *d* = 0.21, BF_01_ = 3.17, 95% credible interval [−8.82, 5.01]). For a graphical illustration of the individual variability within the two groups with the CFMT and the FEQ, see Figure 6.

There was no evidence for a differential deficit between the two groups when CFMT performance was contrasted with CCMT performance (*t*_27_ = −0.11, *p* = 0.91; CFMT and group membership correlation, *r* = 0.423; CCMT and group membership correlation, *r* = 0.446; overall CFMT and CCMT correlation, *r* = 0.285, *N* = 30), or when CFMT performance was contrasted with the performance on the object decision task with silhouettes (*t*_27_ = −0.27, *p* = 0.79; CFMT and group membership correlation, *r* = 0.423; object decision and group membership correlation, *r* = 0.474; overall CFMT and object decision correlation, *r* = 0.359, *N* = 30). However, there was evidence of a differential deficit when FEQ performance was contrasted with CCMT performance (*t*_27_ = −2.33, *p* = 0.03; FEQ and group membership correlation, *r* = −0.098; CCMT and group membership correlation, *r* = 0.446; overall FEQ and CCMT correlation, *r* = 0.085, *N* = 30) and when FEQ performance was contrasted with the performance on the object decision task with silhouettes (*t*_27_ = −2.15, *p* = 0.04; FEQ and group membership correlation, *r* = −0.098; object decision and group membership correlation, *r* = 0.474; overall CFMT and object decision correlation, *r* = −0.15, *N* = 30).

##### Interim Discussion

There was anecdotal evidence in favor of the two groups performing differently on the CFMT but moderate evidence for the 0-hypothesis of the FEQ. These findings do not suggest that the Old-DP group was generally more impaired than the New-DP group on the measures of face recognition. Importantly, we also found positive evidence suggesting that differences between the two groups on the CCMT and the object decision task could not be accounted for by the differences in the FEQ performance as a differential deficit was found in both cases. However, the same was not true in the case of the CFMT because the differences between the groups in the object recognition measures were not significantly larger than what would be expected from the group difference with the CFMT (negative evidence). By weighing the negative and positive findings, these results do not support the idea that the Old-DP group was simply more impaired across all tasks. Rather, the results suggest that the deficit in the Old-DP group reflects something that impairs both face and object recognition, whereas the deficit in the New-DP group seems to be much more selective for faces.

## 4. Discussion

While several hundred cases with developmental prosopagnosia (DP) have been reported over the years since its initial description by McConachie [6], it is still not clear what kind of cognitive dysfunction(s) is underlying the disorder or whether it is the same for all cases with DP. Further, in the cases where seemingly selective impairments for faces have been observed, it has not been examined whether differences in visual similarity between the domains could account for the selectivity [11].

To our knowledge, the only study that has examined the influence of visual similarity in DP was an earlier study from our group [8]. In this study, we found that the differences in visual similarity were likely to account for at least some of the face recognition difficulties observed in the group of DPs. Thus, in an experiment where visual similarity was systematically varied, there was only evidence for a disproportional problem with faces in the condition where faces were more visually similar than the contrast category (houses). When visual similarity was kept constant, the category-effect disappeared. One limitation of this study was that it was based on a relatively small sample (n = 9). Moreover, this group (Old-DP group) also showed impairments on other tests of object recognition, although these difficulties were less pronounced than their face recognition problems [8,16]. Given that the sample was modest, and that DP in different individuals may reflect different underlying mechanisms, this prompted us to examine whether similar effects of visual similarity could be found in another independent and larger group (N = 21) of individuals with DP (New-DP group).

The present study failed to replicate the findings reported by Gerlach et al. [8,16]. Consequently, the present findings do not support the notion that impaired face processing in the New-DP group is driven by visual similarity per se (over and above how it affects the performance of the control participants). In fact, the New-DP group performed quite accurately across all conditions in the critical experiment examining the influence of visual similarity and not significantly different from their control group. This stands in contrast to the Old-DP group, who were impaired across all conditions [8,16]. Consequently, from this experiment, it seems that the New-DP group has a more selective deficit in face recognition than the Old-DP group.

To examine this possibility more thoroughly, we subjected the New-DP group to the demanding tests of object recognition that we have previously found the Old-DP group to be impaired with (the CCMT and an object decision task with silhouettes) [8]. In these tasks, the New-DP group not only performed within the range of the control group but also significantly better than the Old-DP group. Moreover, the New-DP group exhibits a classical dissociation between face and object processing using the stringent criteria suggested by Crawford, Garthwaite, and Gray [25], see also [26].

## 5. Conclusions

By (i) ruling out differences in visual similarity as an underlying factor driving the category-effect (face recognition being more impaired than object recognition), (ii) providing stringent evidence for a classical dissociation between the domains of face and object recognition at the group level, and (iii) using demanding tests of object recognition that have revealed impaired object recognition performance in other cases of DP [8], the present findings are perhaps the strongest evidence yet presented for a face-specific deficit in DP.

There is no clear indication in the data that the difference between the current findings and our previous results [8,16] is related to the severity of the face recognition impairment, and thus, the results seem to suggest that DP is a heterogenous disorder [1,2,3,4]. Consequently, while both groups (Old- and New-DPs) show a disproportional impairment for faces relative to objects and report the face recognition problem to have been present throughout their whole life, the face recognition problem in the two groups may result from different mechanisms. This conclusion seems particularly warranted in the present study because the two groups performed the exact same experiments and were classified with DP using the same ‘diagnostic’ tests (the CFMT, the FEQ, and a clinical interview).

We have argued elsewhere that the deficit underlying the perceptual problems shown by the Old-DP group is likely to reflect a delay in the processing of global shape information that affects both object and face recognition, but that this general deficit has a larger impact on face than object recognition due to the high visual similarity of faces [28]. Based on the experiments presented here, we can offer no similar detailed account of the selective face recognition deficit in the New-DP group. That said, the present results do suggest that the deficit in the New-DP group is likely to arise at a higher level in the perceptual processing hierarchy than the deficit in the Old-DP group, a level where face and object recognition seem to diverge [29,30].

## Figures and Tables

**Figure 1 brainsci-14-00107-f001:**
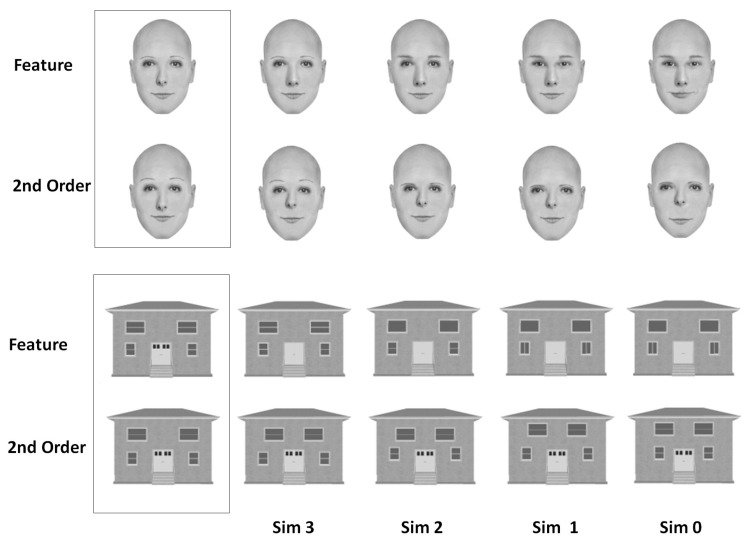
Examples of the face and house stimuli used in the perceptual matching task. ‘Sim’ designates the similarity level with each stimulus differing from the one presented in the boxed area by one difference (Sim 3), two differences (Sim 2), three differences (Sim 1), and four differences (Sim 0).

**Figure 2 brainsci-14-00107-f002:**
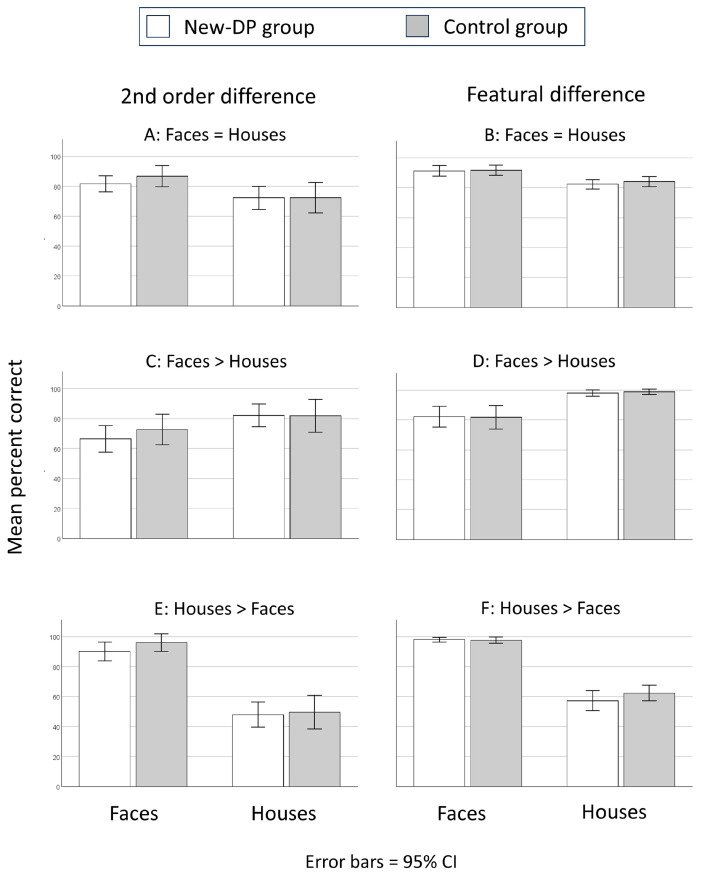
Results from the simultaneous matching task. Performance with faces and houses when the similarity level was equal for the two categories for (**A**) second-order differences and (**B**) featural differences. Performance with faces and houses when the similarity level was higher for faces (Sim 3: see Figure 1) than for houses (Sim 0: see Figure 1) for (**C**) second-order differences and (**D**) featural differences. Performance with faces and houses when the similarity level was higher for houses (Sim 3: see Figure 1) than for faces (Sim 0: see Figure 1) for (**E**) second-order differences and (**F**) featural differences.

**Figure 3 brainsci-14-00107-f003:**
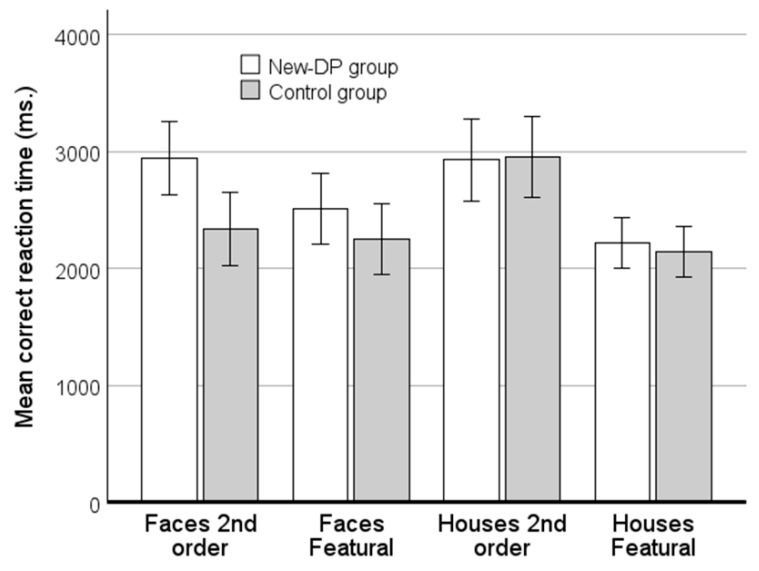
The mean correct RTs for the New-DP group and its control group in the four conditions of the simultaneous matching paradigm with faces and house. It is evident that the New-DPs only differed significantly from the controls in the second-order conditions with faces. For the three other conditions, there was a clear overlap in their associated 95% CIs.

**Figure 4 brainsci-14-00107-f004:**
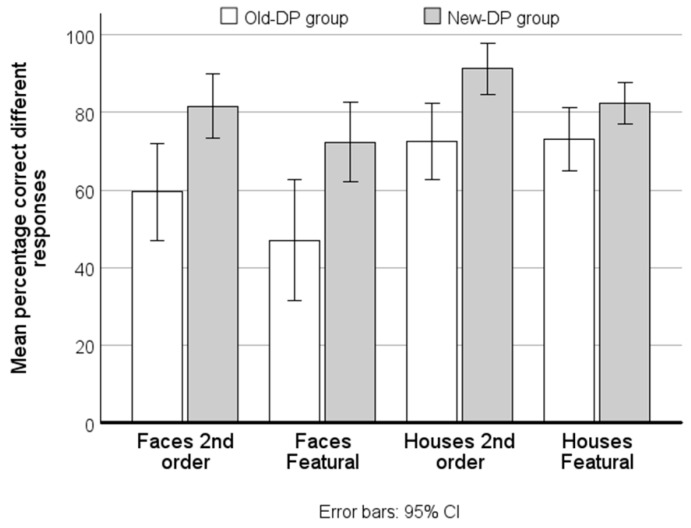
The mean percentage correct across the four similarity levels of the simultaneous matching paradigm for the Old-DP and the New-DP groups. In all conditions the New-DP group is more accurate than the Old-DP group.

**Figure 5 brainsci-14-00107-f005:**
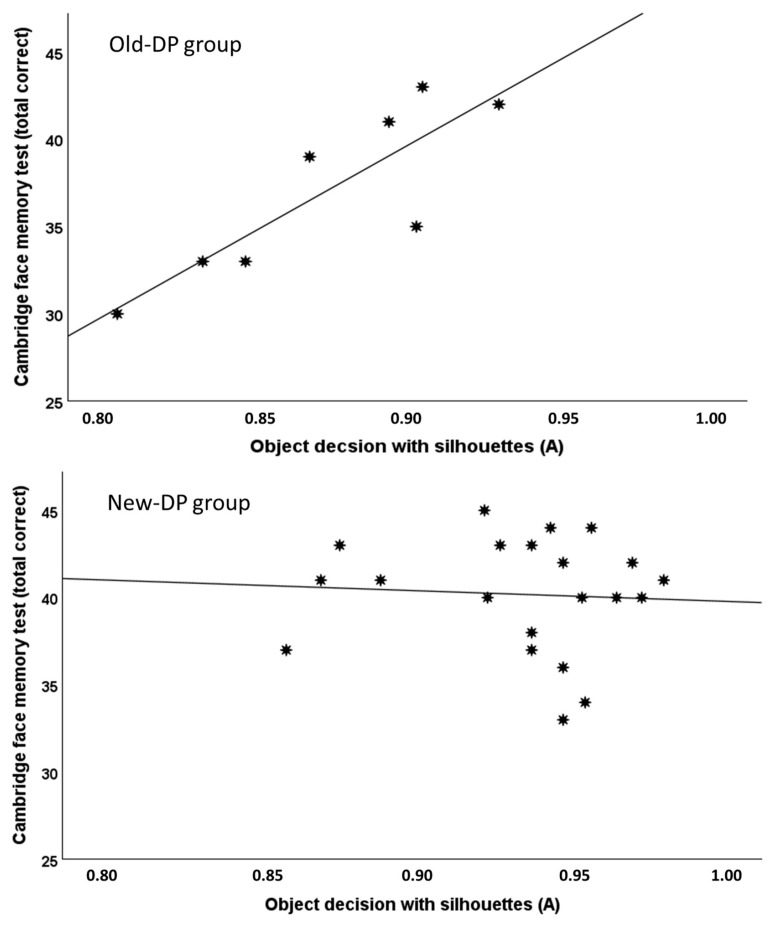
Scatterplots showing the correlation between the object decision task with silhouettes and the Cambridge face memory task for the Old-DP group (**top**) and the New-DP group (**bottom**).

**Figure 6 brainsci-14-00107-f006:**
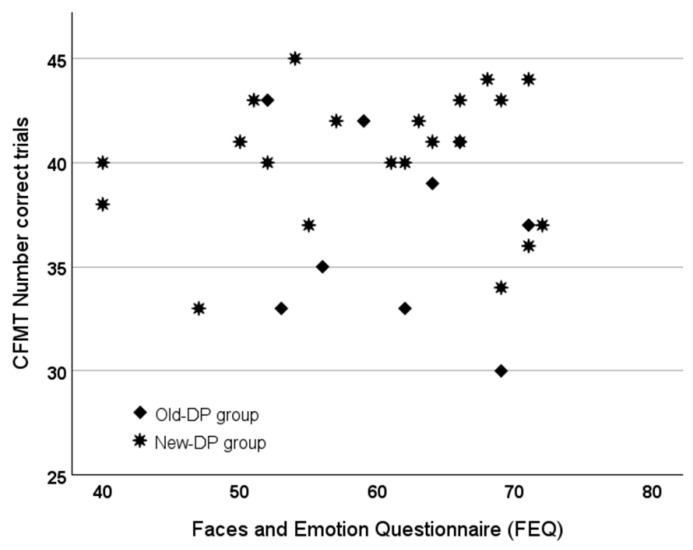
The score obtained on the FEQ (the higher the score, the larger the degree of face recognition difficulty experienced) and the number of correct trials obtained on the CFMT for the members of the Old-DP and the New-DP groups.

## Data Availability

The authors confirm that the data supporting the findings of this study are available at https://osf.io/zv6j9/.

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
