# Peer review of "Evidence for a Classical Dissociation between Face and Object Recognition in Developmental Prosopagnosia"

_brainsci, 2024, doi:10.3390/brainsci14010107_

Round 1

Reviewer 1 Report

Comments and Suggestions for Authors

This research examines the cognitive deficits associated with developmental prosopagnosia (DP) and scrutinizes the variance in face recognition impairments among distinct DP subgroups. The study's significance lies in unraveling variations in cognitive deficits, shedding light on the heterogeneous nature of DP, and delineating its implications for both face and object recognition.
This study aims to analyze and compare face recognition impairments within different DP groups—Old-DP and New-DP—while contrasting their performance with control groups. By elucidating potential differences in cognitive deficits, the research seeks to discern selective impairments in face versus object recognition, aiming to refine our understanding of DP subtypes.
The study employs a battery of tests, including the simultaneous matching paradigm and object recognition tasks, proficiently evaluating face and object recognition abilities across varying similarity levels. The utilization of the Cambridge Face Memory Test (CFMT) and other cogent cognitive assessments significantly bolsters the robustness of the experimental design.
The findings underscore a significant divergence between the Old-DP and New-DP groups. Intriguingly, the New-DP group did not manifest the anticipated severe face recognition impairments observed in the Old-DP group, challenging established assumptions. Moreover, the analyses of object recognition tasks revealed a more discernible deficit in face processing for the New-DP group.
The study contributes substantively to the evolving understanding of DP by highlighting potential variations in cognitive deficits within distinct DP cohorts. The divergence observed between the Old-DP and New-DP groups implies a more nuanced manifestation of face recognition impairments, emphasizing the necessity for a refined characterization of DP subtypes.

My minor Comments for Article Revision:

  1. It would be beneficial to provide more explicit demographic characteristics of the New-DP group, including pertinent details, to enhance the comprehensive understanding of the sample.
  2. Consider furnishing additional contextual information regarding the criteria used for selecting the control group, thereby fortifying the comparability between the control and DP cohorts.

Author Response

We thank Reviewer 1 for the positive evaluation of our manuscript.

Response to Reviewer 1’s points 1 and 2:

In the paper we list the mean/variation for the typical demographic variables: age, gender and educational level for both the individuals with DP and their controls. Information on these variables for each individual is also given in the supporting data that accompanies the paper (https://osf.io/zv6j9/). Based on a suggestion from reviewer 2 we have also added the information that the new DP and control groups were independent from the ‘old’ sample, and we also add that all participants were Caucasian Danish citizens. The full description of demographics on p. 3-4 now reads:

“The individuals in the New-DP group (N = 21) were selected from a larger group of self-referred individuals we have tested over a period of several years. The sample is independent from the Old-DP group, as it was recruited at a later time. Participants for this study were selected based on the criterion that they had completed the simultaneous matching paradigm.

All participants with DP completed structured interviews regarding everyday difficulty with facial identity recognition and possible family history of DP. They all reported severe difficulties with face recognition in their everyday life, as evaluated by the first part of the Faces and Emotion Questionnaire (FEQ, 29-items) [17]. The DP classification was ultimately based on abnormal scores on both the FEQ and the CFMT [18] [for more details see 19]. All DPs performed within the normal range (score of 32 or less) on The Autism-Spectrum Quotient [20], and did not receive remuneration for their participation.

The control group (N = 21) were selected from a larger group of participants who had completed the same test protocol as the DPs including the CFMT and FEQ. They were selected to match one of the New-DPs as closely as possible on gender, age, and education. No controls from the previous study were included, i.e. the sample was independent. All controls performed within the normal range (not below 2 SDs) on the CFMT, evaluated by the age and gender adjusted norms provided in Bowles, et al. [21]. They also performed within the normal range on the Autism-Spectrum Quotient. Controls received gift certificates of ~120 DKK (~20 USD) per hour for their participation.

The New-DP group consisted of 15 females/6 males (age: M = 40, SD = 14; years of education M = 16, SD = 1), and the control group consisted of 15 females/6 males (age: M = 40, SD = 13; years of education M = 16, SD = 1.1). All participants were Caucasian Danish citizens.”

Reviewer 2 Report

Comments and Suggestions for Authors

I found the topic of the manuscript quite interesting and the manuscript itself written relatively well. There are several places in which the exposition can be improved for conceptual clarity and grammar (less so the latter). I think my biggest concern is with the approach to what seems like a largely post-hoc based analysis path that is driven largely by the authors own speculations that are based on the results of the previous test; when declared significant the tests are often, though not always, only marginally so. There are no multiple comparisons corrections applied (at least none reported) and the number of tests seems like overkill to me, particularly those performed on the comparison between the old and new DP groups. It seems reasonable to test the baseline average scores on the various tests, but I'm not sure how informative some of the correlational analysis is. The deficit in reaction times found for 2nd order face differences in the new DP group seems really undersold to me, particularly since this finding firmly rejects the null hypothesis (that the DPs are only disproportionately worse at face discrimination because differences among faces require more fine-grained discrimination than differences among common objects). I think there is also a general reluctance in the community to run correlations with what is essentially a dichotic splits of larger groups. Here, the authors split the overall DP group into ‘old’ and ‘new’, which is understandable due to the difference in time, I guess, between their previous effort and the current one, but the old group is quite small. Specifically, the authors conduct correlations between the face memory test and discrimination sensitivity, A, for object silhouettes for the old and new DP groups, separately. But the old DP group was very small for running a correlation (N=8 points in the upper panel of Fig. 5) – too small for a correlation coefficient to be considered all that reliable. Moreover, the correlations themselves are not compared (is the r value significantly smaller for the new group than the old? -- this is the more important question to resolve statistically than showing that one group's correlation is significant while the other is not). I would have thought that the better approach would be to simply run the correlation on the pooled group to determine if there is a relationship there or not.

In any case, I admit to being in a state of flux with respect to the paper. The topic as mentioned is highly interesting, but I think the authors opted for an awkward approach by favouring the null hypothesis at the outset: that face recognition deficits appear to be unique not because faces are treated as a unique domain of visual processing but because they are more difficult to differentiate than other objects. As the authors admit, the crux of their position relies on negative evidence. They also admit to being a bit self-conscious of this fact, and so they propose additional hypotheses that don’t exactly address the central question but have more to do with the not-so-surprising idea that the more difficult the discrimination task, within a given domain (discrimination between faces and discrimination between houses are tested separately), the poorer the participants’ performance. I think the idea of manipulating discrimination difficulty by manipulating the stimulus parameters is excellent, but I have some concerns that the level of control required to pull off balanced between-domain comparisons is not there. I hesitate to recommend publication at this stage. My specific comments are as follows:

Abstract:

is the larger sample of developmental prosopagnosics independent of the original? Reading on, I believe this is so, but it would be good to mention this in the abstract, because it seems to me a feature of the study the authors can highlight upfront.

The references to an effect of visual similarity in the abstract are a little opaque. The reader would have no idea what this refers to without reading the manuscript, and it seems like a central feature of the abstract itself. Perhaps there is a way to explain what the phrase refers to in the abstract (or rephrase it altogether). From my understanding, this effect is related to the idea that faces are more similar than objects -- although this would depend on the set of objects used wouldn't it? But it could also refer to the effect of the stimuli similarity level, meaning that as the similarity increases, the discrimination task is more difficult. On top of this, I guess the authors could also be referring to a DP vs. control group difference. There are a lot of moving parts, and I don’t think the phrase lends much help disambiguating the possible meanings.

The conclusion in the abstract concerning the "strongest evidence...for a face specific deficit..." is not supported in the abstract by a preceding statement of results. For me at least, this was a little confusing because the authors' statement of findings seem to reference only a failure to replicate a visual similarity effect from the previous study.

Introduction:

The authors cite one of Crawford's articles in which he proposed a more robust definition for single-case study dissociations. Was there any reason the authors preferred the older definitions from Shalice over Crawford's? Crawford included what he referred to as a strong differential dissociation, in which performance on task A was significantly different from performance on task B, regardless of whether performance on each task was significantly worse than controls. What matters in Crawford's definition is the significant performance difference between two tasks relative to controls.

One other possibility for a dissociation between face and object recognition that would fall under the null hypothesis umbrella, I suppose, would be that face recognition is more challenging than object recognition; that the set of face stimuli typically used requires finer-grained discrimination than the set of object stimuli. The challenge then is to ensure the stimuli across the two domains are equally as challenging. I think an argument such as this might fall under the second of the two possible explanations proposed by the authors (the argument about a single, common recognition process or mechanism). As I read on, it seems like the authors do adopt a similar, if not identical, discrimination difficulty based explanation as their preferred hypothesis. I would suggest making the predictions a little more concrete where they can be. For example, the authors write "Considered together, these findings suggests that differences in visual similarity could very well explain the category-effects we observed in this group of DPs." I'm guessing the authors do not merely mean "differences". They mean that the higher visual similarity among faces than for objects is more challenging to discriminate and therefore more likely to yield poorer performance among a population with discrimination deficits.

"The pattern of results was rather clearcut: By manipulating the degree of visual similarity between the categories we could induce a problem that was either more pronounced for faces than for houses (when visual similarity was highest for faces), more pronounced for houses than for faces (when visual similarity was highest for houses), or equally pronounced for both categories (when visual similarity was on par)..."

This could be re-worded for clarity...what do the authors mean by "problem"? The discrimination difficulty? If so, then is that the "sim" differences? By "category", it isn't clear if the authors mean "domain" (faces vs. houses) or if they mean feature vs. 2nd order differences or the sim differences. Sticking with "domain" when referring to faces vs. objects/houses would help.  Also, I would suggest replacing "results" with something framed around task difficulty. I thought the authors manipulated discrimination difficulty within each stimulus domain by two variables, what I’d call ‘whole feature substitution’ vs. 2nd order feature changes. I would think, a priori, that the 2nd order differences would be subtler and therefore more difficult to detect, but the authors do not express an opinion on that front.

"To anticipate the results, we were unable to replicate the effects of visual similarity in the New-DP group." It isn't clear what exactly is meant by the effect of visual similarity. Do the authors mean a between-domain difference in discrimination performance?

"These findings suggest that a classical dissociation can be found between face and object recognition in DP even when object recognition is tested in quite demanding conditions and when effects of visual similarity are unlikely to account for the domain differences." Is the reference to a classical dissociation directed at the old DP group?

Methods

Participants for the DP new group "...were selected based on the criterion that they had completed the simultaneous matching paradigm.". Is this not misleading? It seems like the participants were selected following a number of different tests, were they not? They were not selected merely by the fact that they complete the main task (which might come across as a bit circular).

I wasn’t comfortable with the level of detail provided for the discrimination task. The authors reference their previous paper, but surely there are some key pieces of information about presentation time, for example, decision time limits, etc. that can be reported here.

Can the authors explain how they determined how much they alter the stimuli for the second order manipulations within each domain (e.g., how much extra eye-spacing to introduce)? Was this based on prior piloting? My question stems from concern about a potentially important and unacknowledged assumption that the null hypothesis relies on, which is that the discrimination difficulty at a given similarity level for the two domains, faces and houses, is equal. If instead, they are not equal and the faces at sim level, say, 2, are more challenging to discriminate for the controls than the houses at sim level 2, then it would hardly be surprising to find that a deficit in object discrimination (i.e., the null hypothesis’s view of the DP groups) exacerbates this performance difference in domain. Wouldn’t the best way to ensure equal task difficulty across domains at a given ‘similarity difference’ level be to find the presentation parameters that yield identical discrimination performance across domains for controls such that the discrimination performance varies systematically with ‘similarity level’ only (and not domain)? With the baseline difficulty settled, testing the DP group would be free from any concerns about domain-differences in stimulus discrimination difficulty.

I think the authors did a good job showing that the ‘similarity differences’ within a given domain reflect differences in discrimination difficulty within that domain. But (and echoing my previous comment) I wasn't convinced, based on what is written before the results section, that the discrimination difficulty is equivalent between domains at a given ‘similarity’ level. How do we know that a sim level of 1, the baseline discrimination difficulty for houses and faces is the same? To use a deliberately extreme example, suppose I vary only the distance between the eyes of the set of faces (a 2nd order manipulation) ...if I make the difference in eye-distance large enough, surely the difference between the two faces becomes quite easy to detect -- perhaps even easier to detect than more subtle second-order differences I could introduce across the other features, and yet the sim difference level scheme would suggest the opposite degree of difficulty. Hence my first question about how these second order variations were determined. The same can be said for the feature variations. Ideally, to ensure that comparisons between domains are balanced for discrimination difficulty, the discrimination performance between domains would be shown to be equivalent for the controls (e.g., average discrimination performance for the houses is statistically identical to the average discrimination performance for the faces) by finding the stimulus parameters that achieve such control. That way we’d know we would be working with the stimulus parameters that allow us to start from the same baseline level of discrimination difficulty for both domains, and any differences between the controls and DP group that are specific to the face domain can then be safely be attributed to DP and DP only (and not domain-differences in discrimination difficulty). Absent such an a priori established performance-based control over the discrimination difficulty for the stimulus parameters, we rely on a complex interaction among similarity level, feature manipulation type (feature vs. 2nd order), and domain (faces vs. houses) and remain subject to concerns over the underlying function relating stimulus discrimination difficulty to discrimination performance and how a common discrimination system alters that function in a way that leads to a spurious dissociation.

Was reaction time recorded? If so, it would be meaningful to analyze, given the general concern for a trade-off between processing speed and accuracy. I see that later on in the Results section, the reader encounters an analysis of RT for this exact concern. I suggest referring to the RT analysis in the Methods, because the reader will probably think about this measure, as I have, at this juncture of the manuscript and wonder why it is missing. Moreover, the prediction would be that the DP group is disproportionately slower to discriminate faces than houses, relative to the controls, unless the domains are matched for discrimination difficulty. While on the topic of RT, the authors could consider combining accuracy and RT to compute an efficiency score (e.g., Townsend, J.T., Ashby, F.G., 1983. Stochastic Modeling of Elementary Psychological Processes). One simple way to do this would be to divide reaction time by accuracy (proportion correct).

Results

The manuscript refers to domain, category and now "type" to refer to faces vs. objects/houses. I would suggest maintaining "domain" throughout the manuscript.

Why is the mean square error (guessing that is what "MSe" stands for in the manuscript) reported?

Figure 2 is a little confusing. What does the panel title "Faces = Houses" mean? (and the other panel titles). Are they reference to the "similarity" levels? Why not simply use line graphs with four series: two for the houses and two for the faces; two for the featural differences and two for the 2nd order ones, with ‘similarity’ level on the x-axis? A line graph would be appropriate because the similarity levels, at least in theory, were designed to capture a quantitative measure, 'discrimination difficulty', were they not? Show the reader the means and SEs for the conditions for both accuracy, RT, and efficiency.

The authors note a "significant difference" between the DPs and controls in terms of their RTs for the 2nd order differences in faces, but the direction of the difference depicted in Figure 3 is important. I would suggest reporting the direction of the difference…which is slower RTs for the DP vs. controls when discriminating faces. Doesn’t this finding reject the null hypothesis? Doesn’t this suggest the DPs have a domain-specific deficit? In the interim discussion concerning this section of the results, it seems like the authors believe otherwise (see next comment).

What is a sequential matching strategy? This phrase is introduced in the subsection 3.2.1. Interim discussion of the Results section with no explanation and is not mentioned anywhere else in the manuscript. Furthermore, in this same paragraph the authors express concern that the DP group performed well in task despite their CFMT and FEQ indications. This seems to me to ignore the fact that the DP group was significantly slower than the controls at discriminating faces with 2nd order differences ("This revealed that the New-DP group and the controls differed significantly in the condition with matching of 2nd order relations for faces (t40 = 2.77, d = .85, p < .01)"...as noted above, the direction of the difference is that the DP group is slower). One would think that the 2nd order differences would be the more difficult of the two difference types (feature vs. 2nd order) to discriminate. In other words, if we are to take the manipulations at face value, the tests of 2nd order differences seem to me to be the most sensitive to detect group differences, no?

The authors analyze the accuracy across the old and new DP groups, but they do not test for RT differences. Is this not possible? If the RT data exist, it should be subjected to the same analysis, and the authors should consider computing an efficiency score from combining the accuracy and RT data and performing a similar analysis.

The subheadings could use some revision for clarity, and I wonder if the results section could be significantly streamlined.

Comments on the Quality of English Language

Seems mostly ok to me. There are some places that need clarification and other areas where revision is required to correct grammatical problems.
